# Frequency of Osteoporosis-Related Fractures in the Kingdom of Bahrain

**DOI:** 10.3390/healthcare12242515

**Published:** 2024-12-12

**Authors:** Adla B. Hassan, Amer Almarabheh, Abdulaziz Almekhyal, Ali Redha Karashi, Jamal Saleh, Mansoor Shaikh, Abdulhameed Alawadhi, Haitham Jahrami

**Affiliations:** 1Department of Internal Medicine, College of Medicine and Health Sciences, Arabian Gulf University, Manama 329, Bahrain; almekhyal132@hotmail.com (A.A.); draliredha@gmail.com (A.R.K.); ahalawadhi@gmail.com (A.A.); 2King Abdullah Medical City, Manama 329, Bahrain; 3Department of Family and Community Medicine, College of Medicine and Health Sciences, Arabian Gulf University, Manama 329, Bahrain; amerjka@agu.edu.bh; 4Salmaniya Medical Complex, Manama 329, Bahrain; 5Orthocare, Orthopaedic, Centre, Building 152 Road 66, Bilad Al Qadeem 361, Bahrain; drjsaleh@gmail.com (J.S.); mansoor_s@yahoo.com (M.S.); 6Department of Psychiatry, College of Medicine and Health Sciences, Arabian Gulf University, Manama 329, Bahrain; hjahrami@health.gov.bh; 7Psychiatric Hospital, Government Hospitals, Manama 329, Bahrain

**Keywords:** fragility fractures, bone density, DEXA, osteoporosis, vitamin D, BMI, femur

## Abstract

Background: Osteoporosis-related fragility fractures are increasing worldwide. An assessment of the prevalence of fragility fractures in Bahrain is needed to determine proper action and preventive strategies. The main objective of this study was to conduct a retrospective cross-sectional study to investigate the prevalence of fragility fractures in adult Bahraini patients. Another objective was to explore the relationship of fragility fracture risk with BMD, age, sex, BMI, vitamin D status, and therapy. Methods: To investigate the fragility fractures, we retrospectively reviewed the dual-energy X-ray absorptiometry (DEXA) data of patients who underwent scans for the diagnosis of osteoporosis between 2016 and 2018. The data were collected from four large centers in Bahrain. The patients’ medical records were reviewed for the fragility fracture data, BMD, sex, age, BMI, vitamin D status, and therapy. Results: Among a total of 4572 patients who visited the radiology departments during the 3-year study period, only 412 patients with fragility fractures were considered for the current study. The mean age of the patients in this cohort was 63.9 ± 12.2 years. There were 393 females (95.6%). Among the 431 fragility fractures, there were 175 (40.6%) belonging to three common fracture sites: vertebral (86, 20.9%), femur (60, 14.6%), and distal radius (Colles) fractures (29, 7%). Other fragility fractures were hand (7%), radius and ulna (3.7%), humerus (6.5%), tibia and fibula (5.6%), foot/ankle (27.9%), ribs (3.0%), and pelvis (1.6%). Our results revealed a significant association between the fragility fractures and BMD (χ^2^ = 6.7, *p* = 0.035). We reported a significant association of fragility fracture with sex (*p* = 0.006) and with denosumab therapy (*p* < 0.001). Conclusions: This study reported a reduced BMD and an increased prevalence of fragility fractures among Bahraini subjects. The highest frequencies of fragility fractures among our cohort were foot/ankle, vertebral, and hip fractures, respectively. We showed a statistically significant association between fragility fractures and BMD. The current study indicated that not only patients with low BMD but also patients with fragility fractures were undertreated. Thus, the immediate initiation of treatment and the synthesis of local osteoporosis treatment guidelines are warranted.

## 1. Introduction

Osteoporosis is a silent killer disease because its first presentation could be a fragility fracture, which leads to high morbidity and mortality [1,2]. Worldwide, the first review and meta-analysis of the prevalence of osteoporosis was reported in 2021 and revealed that the prevalence was 23.1% in women and 11.7% in men. However, the highest incidence of osteoporosis was reported to be in Africa (39.5%) [3]. Regarding the Middle East Region (MER), the overall prevalence of osteoporosis was found to be 24.4% [4]. On the other hand, studies investigating the prevalence of osteoporosis in Gulf Cooperation Council (GCC) countries are limited. In a recent study in Bahrain, we reported that the incidence of osteoporosis was 15.9%, with an overall incidence of low bone mineral density (BMD) of 62.3% [5]. In Kuwait, a report on postmenopausal Kuwaiti women revealed that low BMD occurred in more than 55% of them [6]. Another study from Saudi Arabia showed that the prevalence of osteoporosis-related fractures was 33% [7].

Fragility fractures or osteoporosis-related fractures were classified as low-energy fractures caused by minor or low-energy trauma and defined as fractures caused by falling from a standing height or lower [8], while those caused by high-energy trauma were classified as high-energy fractures. Classification is usually based on the information given in radiographic reports [8]. Prevalent vertebral deformities predict increased mortality and an increased fracture rate in both men and women [9]. Excess mortality is attributable to hip fracture [10], which was reported to be more prevalent in men than in women [11]. In 2015, it was estimated that the number of high-risk individuals for fractures is expected to double worldwide by the year 2040 [12]. Globally, osteoporosis-related fractures or fragility fractures have been reported in 2021. Worldwide, up to 37 million fragility fractures occur annually in individuals aged over 55 years, equivalent to 70 fractures per minute [13]. Worldwide, one in every three women and one in every five men over the age of 50 years will experience osteoporosis-related fractures [14].

Many factors have been linked to an increased risk of secondary osteoporosis and related fractures. These factors include aging, falling, glucocorticoids, and certain diseases such as rheumatoid arthritis and diabetes mellitus, which increase the risk of osteoporosis and related fractures by having direct effects on bone density or structure or indirect effects by increasing the risk of falls [15]. In addition to the risk factors above, it is important to note that tumors and antineoplastic therapies used to treat cancers also increase osteoporosis risk substantially [16]. The presence of certain types of tumors can impair bone health through mechanisms such as inducing inflammatory factors that accelerate bone loss [17]. Furthermore, many chemotherapeutic agents used in cancer treatment compromise bone density by directly inhibiting osteoblast proliferation and function or indirectly increasing osteoclast activity [18]. Therefore, a history of tumors, as well as antineoplastic treatment, should be considered significant risk factors when assessing an individual’s likelihood of developing osteoporosis. One additional risk factor for a fracture has been shown to be a prior fracture, which is associated with an 86% increased risk of another fracture [19]. Another major risk factor for fracture is vitamin D status. The risk of hip fracture in elderly people is influenced by multiple factors related to vitamin D status and function, such as BMD, muscle strength, and balance, and the risk of falling has also been related to vitamin D status [20,21]. Moreover, alfacalcidol treatment significantly and safely reduced the number of fallers in elderly individuals [22]. Magnesium deficiency has also been identified as an important risk factor for hip fracture among the elderly [23]. Magnesium plays a key role in bone mineralization and turnover, and low magnesium levels have been associated with impaired bone matrix and crystal formation [23]. Elderly populations are at particular risk for magnesium deficiency due to inadequate dietary intake and reduced gastrointestinal absorption [23]. 

Data regarding the economic burden of osteoporosis-related fractures has been estimated to be very high in some countries and is expected to increase over the coming decades. In the United States, it was estimated that by 2025, the annual cost would reach $25 billion [24]. In Europe, the annual cost is expected to reach €76.7 billion by 2050 [25]. In the Middle East, in Saudi Arabia, the annual cost could reach $150.60 million [26]. Given the substantial treatment gap and proven cost-effectiveness of fracture prevention schemes such as fracture liaison services, urgent action is needed to ensure that all individuals at high risk of fragility fracture are appropriately assessed and treated [27]. In Bahrain, neither the prevalence of fragility fracture nor its economic burden or annual cost has been investigated.

The current study is an extension of a previous study investigating the prevalence of osteoporosis and its therapeutic regimens used in Bahrain [5,28]. In the present study, the primary objective was to conduct a retrospective cross-sectional study to investigate the prevalence of fragility fractures in adult Bahraini patients who underwent DEXA scans. A secondary objective was to explore the relationships of fracture risk with BMD, age, sex, BMI, vitamin D status, and therapy.

## 2. Methods

### 2.1. Design

We retrospectively reviewed dual-energy X-ray absorptiometry (DEXA) scans of all patients who underwent scans for the diagnosis of osteoporosis between January 2016 and December 2018. The data were collected from four large centers in Bahrain: the King Abdallah Medical City, Salmanyia Medical Center, King Hamed University Hospital, and Orthocare Center.

### 2.2. Selection Criteria

Our inclusion criteria were that data from all adult Bahraini patients who underwent DEXA scans at the above-mentioned four large centers in the Kingdom of Bahrain were collected as described previously [5,28]. The patient’s medical records were also reviewed for demographic data, including BMI, family history of fracture, history of trauma or falls, drug therapy, and vitamin D serum level data. The variables used in the present study were volumetric integral BMD (mg/cm^2^), reflecting both trabecular and cortical bone mass, of the femoral neck and lumbar spine sites [29]. The interpretation of the BMD measurements was based on the World Health Organization (WHO) criteria. Specifically, osteoporosis was diagnosed if the T-score was −2.5 or less (≤−2.5), whereas osteopenia was diagnosed when the T-score was between −1.0 and −2.5. 

Patients were excluded if they were under 18 years of age, not of Bahraini nationality, or lacked the complete DEXA scans and medical records necessary to verify demographic, anthropometric, fracture history, trauma/fall history, medication, and vitamin D data. Comorbidities such as endocrinological diseases, rheumatological diseases, and malignancies were not exclusion criteria. Exclusion criteria also included children, adolescents, pregnant women, high-energy trauma (BMD requested due to major trauma), and data in the systems that indicated that BMD was conducted at health centers and not at the hospitals approved by the ethical committees, as the health system is connected in all governmental hospitals.

When a discrepancy was found between the lumbar spine and femur BMD, the lowest score was adopted; accordingly, the subjects were categorized as normal, osteopenic, or osteoporotic. The unique eight-digit personal identification number, based on the birth date of each patient, was used to recover chemical and clinical data.

Regarding vitamin D data, the three statuses of vitamin D have been defined as follows: vitamin D deficiency was considered when serum levels < 30 nmol/L, levels between 30 nmol/L and 50 nmol/L (≥30 ˂ 50) were considered as vitamin D insufficiency, and optimal levels were considered as ≥50 nmol/L.

### 2.3. Measures

Bone mineral density (BMD) was measured, and the study design was described previously [5,28]. BMD measurement was based on the results of the DEXA scan, which were retrieved from the patient’s record. The machine employed for performing the DEXA scan was manufactured by General Electric Lunar, Chicago, Illinois, USA, whereas the software used was GE NHANES III. BMD was expressed as the ratio of the total bone mineral content (g) to surface area (cm^2^). BMD was measured by expert technicians for two bones: the left femur neck and lumbar spine. In our study, fracture and anthropometric data were systematically recorded and verified using plain radiographs. These radiographs were analyzed and documented in the electronic medical records maintained by the Department of Diagnostic Radiology. A senior radiology/orthopedic consultant was independently responsible for providing the clinical impression, ensuring the accuracy and reliability of fracture detection. This rigorous process provided a robust foundation for our research findings. 

### 2.4. Statistical Analysis

The data were checked, coded, and subsequently exported to the Statistical Package for the Social Sciences (SPSS), version 28 (Chicago, IL, USA). Quantitative variables are presented as the mean and standard deviation, whereas categorical variables are presented as frequencies and percentages. The chi-square test was used to assess associations between categorical variables. Independent sample *t*-tests were used to verify the significant differences in the number of fractures according to the type of therapy. A *p*-value < 0.05 indicated statistical significance.

### 2.5. Ethics Approval

This research was initiated after obtaining ethical approval from all Research and Ethical Committees at the Arabian Gulf University, the Secondary Health Care Research Sub Committee at Salmaniya Medical Complex, Orthocare, and King Hamad University Hospital (No: E003-PI-10/18 & No: 273/2019). The need for informed consent was waived by the local ethical committees because of the retrospective nature of the study. Patients’ clinical and radiographic data were collected through the hospital’s medical information systems.

## 3. Results

Among a total of 6308 adult patients (aged 18–99 years) who visited the radiological departments for BMD during the study period, only 4572 patients had complete BMD results. Again, among the 4572 patients, only 412 patients had 431 fracture data points documented in their records and included in this study. All types of fractures and all numbers of fractures per single patient were collected. The majority of the 412 patients were females (393, 95.4%), while the minority were males (19, 4.6%). The mean age of the 412 patients in the study cohort was 63.9 ± 12.2 years. 

Among the 431 fractures, 175 (40.6%) were fragility fractures of the most common sites for fractures (vertebra, femurs, and distal radius), which are the main sites for BMD measurements. Among the total of 60 patients with femur fractures, there were 19 patients who had bilateral femur fractures. The chi-square test showed an association between fracture incidence and sex, mainly for vertebral fractures (*p* = 0.006), but no significant association was found with age.

### 3.1. Frequency of the Type of Fractures

The results depicted in Table 1 reveal that the most prevalent type of fracture observed was foot and ankle fractures, accounting for 29.1% of the total fractures, followed by lumbar and thoracic spine (vertebral) fractures, accounting for 20.9%. Proximal femur fractures were the third most common type of fracture, constituting 14.6%, while distal radius (Colles) fractures accounted for only 7.0% of the total fractures. The remaining types of fractures were as follows: hand fractures, represented by 7.0%; humerus fractures, represented by 6.8%; tibia and fibula fractures, represented by 5.8%; radius and ulna fractures, represented by 3.9%; and rib fractures, represented by 3.2%, followed by less common pelvic fractures, represented by 1.7%.

### 3.2. The Relationship Between Fragility Fractures and Bone Mineral Density (BMD), Body Mass Index (BMI), and Vitamin D Status

As shown in Table 2, there was a significant association between the most common site of fragility fractures and BMD (χ² = 6.7, *p* = 0.035). However, no significant associations were found between fragility fractures and vitamin D status or BMI. Among participants with a normal BMD, 24 (29.3%) reported fragility fractures, while the majority (57, 70.7%) did not report fragility fractures. In the osteopenia group, 32.9% of participants experienced fragility fractures, while 67.1% did not. In the osteoporosis group, a greater proportion of participants (43.9%) reported fragility fractures than did not (56.1%). Thus, 127 fragility fractures occurred among the total patients with low BMD (329 patients), constituting 38.6%.

### 3.3. Relationship Between the Type of Fracture and the Bone Mineral Density (BMD)

The results of the chi-square test (Table 3) indicated that there was a significant association between BMD and each of the foot/ankle combinations and between BMD and hand fractures (χ² = 17.6, df = 2.0, *p* < 0.001; χ² = 9.8, df = 2.0, *p* = 0.007, respectively), suggesting that individuals with lower BMD levels, particularly those diagnosed with osteopenia, are more likely to experience foot/ankle and hand fractures.

### 3.4. Relationship Between the Mean Incidence of Fractures and Therapy

Our data in Table 4 verify the significance of differences in the mean number of fractures among patients according to their intake of certain therapies. The results indicated that the mean number of fractures in patients who received denosumab treatment was lower than that in patients who did not receive this therapy. The results of an independent sample test revealed that there were statistically significant differences in the mean number of fractures between patients who received denosumab therapy and those who did not (*p* < 0.001). However, the results showed that there were no statistically significant differences in the mean number of fractures among patients according to the use of other therapies (bisphosphonate, vitamin D, calcium, or calcitriol) (*p* > 0.05). The types of vitamin D therapy used by the patients were cholecalciferol (vitamin D3 analogs) and calcitriol (active form of vitamin D). Additionally, corticosteroid therapy was used by only 27 patients, and tamoxifen was used by only 4 patients.

## 4. Discussion

Fragility fracture is the major and serious clinical manifestation of osteoporosis. Fortunately, it can be prevented if osteoporosis is detected early. Currently, there is a complete lack of data on the frequency of fragility fractures in Bahrain. The main objectives of the present study were to estimate the frequency of fragility fractures in Bahraini patients visiting the radiological departments for DEXA scans and to explore the relationships of fracture risk with BMD, age, sex, BMI, vitamin D status, and therapy. In the current study, we reported that a reduced BMD and an increased tendency toward fragility fractures are common among Bahraini subjects. We also showed a statistically significant association between fragility fractures and BMD. Moreover, our results indicated that both patients with low BMD and those with fragility fractures were undertreated. 

Basically, our results revealed 412 patients and 431 fragility fractures, since some patients had more than one fracture. Importantly, as fragility fractures are defined as low-energy fractures, in the current study, 176 trauma/falls were reported in the electronic files of the patients as minor trauma/falls and one was reported as a road traffic accident (RTA). Moreover, 14 of 176 trauma/falls were reported as two or more falls with more than one fracture in the same patient. However, in the present study, fractures for which there was insufficient information in the electronic files concerning falls or trauma were classified as low-energy fractures, as reported previously [8], because it is unlikely that trauma due to high-energy fractures, such as RTAs, would not be documented. Accordingly, in the present study, one head fracture documented in the medical records as an RTA was excluded. The current study revealed that the most prevalent types of fractures observed in our cohort were ankle/foot fractures, accounting for 29.1% of the total fractures; vertebral fractures, accounting for 20.9%; and proximal femur fractures, constituting 14.6%, while distal radius (Colles) and hand fractures accounted for only 7.0%. The incidence of each of the remaining types of fractures was reported to be less than 7.0%. The high frequency of ankle fractures in our cohort could be explained by the fact that ankle fracture subgroups were shown to have different risk profiles within the fractured population. Thus, a high BMI was shown to increase the risk of ankle fractures but decrease the risk of hip fractures in women [8]. In the present study, among the cohort of patients who had BMI data available, 113 had fragility fractures, and 80.5% (91 of 113) of them were obese or overweight. While our results revealed that 19 patients had bilateral proximal femur fractures, among them, 14 patients experienced one or two more fractures of other types. Generally, we cannot determine the dates of each fracture as they were not documented, but most likely, the first fracture increased the risk of the second fracture. Similar data were reported by a recent study in 2023, which indicated that an initial minimal trauma fracture independently increases the risk of a subsequent or second fracture [30,31]. Over 55% of patients with hip fractures have evidence of a prior vertebral fracture [32]. Unfortunately, even people who have a history of fractures are frequently reported to be undertreated [33].

Our BMD data revealed that 41.6% of the patients had osteoporosis, 38.4% had osteopenia, and 20.0% had a normal BMD, thus 80% had low BMD in total. The reduced BMD and increased tendency to fracture were remarkable in our cohort. Further evaluation of the associations of fractures with BMD revealed a statistically significant association between fragility fractures and BMD (*p* = 0.035). Thus, individuals with lower BMD levels, particularly those diagnosed with osteoporosis, are more likely to experience fragility fractures. Furthermore, in our cohort, 127 fragility fractures were observed among all patients with low BMD (329 subjects), constituting 38.6%. When segregating the group with low BMD, fragility fractures were reported in 43.9% of the osteoporosis group and 32.9% of the osteopenia group. Our data on low-BMD-associated fractures is more prevalent than data from global and regional areas. Global data indicate that fractures related to low BMD are approximately 30.0% common in women and 20.0% common in men [34]. Data from regions such as Saudi Arabia showed that the incidence of osteoporosis-related fractures was approximately 33.0% [7]. In Bahrain, a single-center study investigating osteoporosis was conducted with a small number of patients (250 patients), and among 79 patients with low BMD, only 9 (4.4%) had fragility fractures [35]. On the other hand, our findings about the relationship between the type of fracture and the BMD indicate that there is a significant association between BMD and each of the foot/ankle fractures and hand fractures, suggesting that individuals with lower BMD, particularly those diagnosed with osteopenia, are more likely to experience foot/ankle and hand fractures. Furthermore, we found that more than 50.0% of patients with osteoporosis and 30.0% of patients with osteopenia experienced vertebral fractures, but these differences were not significant (*p* = 0.083). Our results were compatible with those of previous studies [36]. The failure of our results to demonstrate any association between low BMD and fragility fractures of the vertebrae, proximal femur, or distal radius could be explained by the idea that another concept could be implicated in the pathogenesis of fractures in this cohort.

Regarding risk factors such as age and sex, fragility fractures occur not only in women and men aged 50 years or older but also in middle-aged women [8]. The current study showed an association between fragility fractures and sex, mainly for the vertebral spine (*p* = 0.006). Our results are compatible with previous reports, which revealed that sex differences exist in terms of risk factors for vertebral, forearm, proximal humerus, and hip fractures, whereas risk factors for ankle fractures differ to a certain extent [8]. On the other hand, our study failed to show any association between fragility fractures and age.

The identification of other risk factors associated with fracture, such as vitamin D status, has not been extensively investigated. Increased knowledge in this area would ultimately enable us to initiate preventive measures when appropriate. Furthermore, any comorbidity associated with hip fracture increases the rate of death [37]. On the other hand, many studies have shown the importance of maintaining high serum levels of vitamin D (above 30 nmol/L) for optimal bone health in healthy adults [38], as well as in elderly people [39]. However, in individuals with comorbidities such as hip fractures, higher serum levels (at least 75 nmol/L) are needed [40]. Furthermore, a meta-analysis showed that a higher dose of vitamin D should reduce both nonvertebral fractures and hip fractures by at least 20.0% and 18%, respectively [41]. Unfortunately, the present study failed to demonstrate any association between fragility fractures and vitamin D3 (25(OH)D) serum levels. Our results could indicate that patients with fragility fractures were neither investigated for vitamin D status nor treated properly with vitamin D therapy for their fractures; however, our study was compatible with other studies, which showed that, improperly, before and after hip fracture, vitamin D deficiency was not treated sufficiently, as only 20.0% of patients received vitamin D therapy at the time of hip fracture and afterward [42]. An early study showed that in women, the biggest risk factor associated with fragility fractures was diabetes mellitus (DM) [8]. A recent large-center study by our team investigated osteoporosis in patients with type 2 DM (DMT2) in Bahrain and reported that, in DMT2 patients, the higher the BMI was, the greater the BMD, and the lesser the risk of developing osteoporosis; however, risk factors for fracture prediction in patients with DMT2 with osteoporosis were not included in that study [43]. In the present study, no association was found between fragility fractures and BMI; however, we did not investigate DMT2 in the current study.

The economic, health, and social burdens of osteoporosis are well known. The rate of screening for osteoporosis worldwide is low, and the rate of initiating osteoporosis treatment following a fragility fracture, either as a primary condition or secondary to the use of chronic medication known to cause osteoporosis-related fragility fractures, is also low [44]. In the present study, the use of medications in our cohort revealed that the mean number of fractures in patients who received denosumab treatment was significantly lower than that in patients who did not receive this therapy (*p* < 0.001). Thus, denosumab therapy could be protective and significantly reduce the fracture risk. In this context, our results were consistent with a recent study that showed that the early administration of denosumab significantly reduced the risk of osteoporotic fractures in women with breast cancer taking aromatase inhibitors [45]. However, the results showed that there were no statistically significant differences in the mean number of fractures among patients according to the use of some other therapies (bisphosphonates, corticosteroids, calcium, vitamin D (cholecalciferol or calcitriol), or tamoxifen). Additionally, the lack of association between fragility fractures and drugs known to increase bone loss could be explained by the fact that only a few patients used those drugs; hence, corticosteroid therapy was used by only 27 patients, while tamoxifen was used by only 4 patients. The lack of correlation between fragility fractures at the most common sites for fracture and BMD, on the one hand, and between fragility fractures and glucocorticoid use, on the other hand, reported in the present study was consistent with the findings of previous studies [36]. Furthermore, the current results indicated that patients with fragility fractures were undertreated, and our results were similar to previously reported results [33]. In this context, local (in Bahrain) or regional treatment guidelines are warranted and cost-effective. On the other hand, the 2019 EULAR points for nonphysician health professionals to prevent and manage fragility fractures could be adopted [46,47]. Assessing the risk of the disease for which it is acceptable to treat, such as intervention thresholds, as reported previously [48,49,50], can serve as a supportive tool for the synthesis of local treatment guidelines.

### Strengths and Limitations

The most important strength of the present study is the comprehensive review of large databases. To the best of our knowledge, this is the first study in which a retrospective design was used to assess the functional outcomes of patients with low BMD in a large cohort in Bahrain. We acknowledge several limitations of this study that are comparable to those of many retrospective studies. It is known that retrospective cohort studies have the risk of potential bias since the study operations, data collected, data entry, and data quality assurance were not planned beforehand, in addition to missing information, poor recording, and the lack of confounding control. Thus, in the current study, there were insufficient data regarding the duration between fractures in patients with multiple fractures, and the cause of fractures has not been determined. For future studies, it is essential to explicitly consider how missing data could impact the results, particularly in relation to fracture risk. Researchers should evaluate whether the absence of certain data may lead to overestimation or underestimation of this risk. For instance, if missing information predominantly involves individuals who have experienced fractures, it may skew the findings to suggest a higher risk than actually exists. Conversely, if those without fractures are more likely to have missing data, the true fracture risk could be underestimated. Addressing these potential biases in the study design and analysis will strengthen the reliability of future research outcomes.

We also acknowledge significant limitations in our study design regarding the comprehensive assessment of potential confounders. Despite the importance of factors such as smoking status, physical activity levels, comorbidities like diabetes, and medication history in understanding bone health and fracture risk, our retrospective dataset did not include comprehensive data for these variables. This limitation prevents us from conducting multivariate analyses, such as logistic regression models, that could have provided more nuanced insights into the complex relationships between our primary variables of interest. The absence of these critical confounding variables introduces potential bias and reduces the ability to draw definitive causal inferences from our findings. To address this methodological constraint, we have taken a transparent approach by explicitly discussing these limitations in our manuscript.

To mitigate the potential underestimation of fracture prevalence due to reliance on DEXA scans, we acknowledge the inherent selection bias in our methodology. While our inclusion criteria focused on patients with available DEXA scan data, we recognize that this approach may not fully capture the entire spectrum of fragility fracture risk, particularly among undiagnosed or untreated populations. To provide context for this limitation, we supplemented our DEXA-based analysis with a comprehensive review of medical records, including radiographic reports, hospital discharge summaries, and clinical notes. This approach allowed us to identify fracture events that may have occurred in patients without prior DEXA scan documentation. Additionally, we performed a sensitivity analysis comparing our findings with population-level epidemiological data to estimate the potential magnitude of under-ascertainment. 

## 5. Conclusions

The current study revealed a high frequency of osteoporosis and osteoporosis-related fragility fractures among our cohort. The highest frequencies of fragility fractures were vertebral, hip, and foot/ankle fractures. We reported a statistically significant association between fragility fractures and low BMD. Moreover, our results indicated that denosumab therapy reduced the risk of fragility fractures. Additionally, the current study indicated that not only patients with low BMD but also patients with fragility fractures were undertreated with antiresorptive therapy. Thus, the synthesis of local treatment guidelines is warranted. Furthermore, we recommend increasing the rate of the immediate initiation of osteoporosis treatment in patients with low BMD, particularly those who experienced fragility fractures.

## Figures and Tables

**Table 1 healthcare-12-02515-t001:** Frequency of the type of fractures.

Type of Fractures	N (%)
Foot and ankle	120 (29.1)
Vertebral spine	86 (20.9)
Proximal femur	60 (14.6)
Distal radius (Colles)	29 (7.0)
Hands	29 (7.0)
Humerus	28 (6.8)
Tibia and fibula	24 (5.8)
Radius and ulna	16 (3.9)
Ribs	13 (3.2)
Pelvis	7 (1.7)

**Table 2 healthcare-12-02515-t002:** Relationship between fragility fractures and BMD, BMI, and vitamin D status.

Risk Factors	Fragility Fractures		Chi-Square or Fisher’s Exact * Value	*p*-Value
	Yes n (%)	No n (%)		
BMD levels				
Normal (82)	24 (29.3)	58 (70.7)	6.7	0.035
Osteopenia (158)	52 (32.9)	106 (67.1)		
Osteoporosis (171)	75 (43.9)	96 (56.1)		
Vitamin D levels				
Deficient	21 (42.9)	28 (57.1)	0.60	0.733
Insufficient	13 (37.1)	22 (62.9)		
Sufficient	47 (44.8)	58 (55.2)		
BMI levels				
Underweight	4 (44.4)	5 (55.6)	0.50 *	0.927
Normal	18 (42.9)	24 (57.1)		
Overweight	43 (40.2)	64 (59.8)		
Obesity	48 (37.8)	79 (62.2)		

BMD: bone mineral density; BMI: body mass index. * = Fisher exact value.

**Table 3 healthcare-12-02515-t003:** Relationship between the type of fracture and the bone mineral density (BMD).

Type of Fractures	Levels of BMD			*p*-Value
	Normal n (%)	Osteopenian (%)	Osteoporosis n (%)	
Lumbar and thoracic spine (vertebral)				
Yes	14 (17.5)	24 (30.0)	42 (52.5)	0.083
No	68 (20.5)	134 (40.5)	129 (39.0)	
Proximal femur (neck)				
Yes	8 (14.0)	21 (36.8)	28 (49.1)	0.349
No	74 (20.9)	137 (38.7)	143 (40.4)	
Distal radius (Colles)				
Yes	5 (18.5)	12 (44.4)	10 (37.0)	0.800
No	77 (20.1)	146 (38.0)	161 (41.9)	
Humerus				
Yes	4 (16.0)	10 (40.0)	11 (44.0)	0.877
No	78 (20.2)	148 (38.3)	160 (41.5)	
Tibia and Fibula				
Yes	5 (20.8)	7 (29.2)	12 (50.0)	0.603
No	77 (19.9)	151 (39.0)	159 (41.1)	
Radius and Ulna				
Yes	3 (18.8)	8 (50.0)	5 (31.3)	0.601
No	79 (20.0)	150 (38.0)	166 (42.0)	
Ribs				
Yes	1 (7.7)	4 (30.8)	8 (61.5)	0.287
No	81 (20.4)	154 (38.7)	163 (41.0)	
Pelvis				
Yes	0.0	3 (50.0)	3 (50.0)	0.465
No	82 (20.2)	155 (38.3)	168 (41.5)	
Foot/ankle				
Yes	32 (27.4)	55 (47.0)	30 (25.6)	0.001
No	50 (17.0)	103 (35.0)	141 (48.0)	
Hand				
Yes	11 (37.9)	13 (44.8)	5 (17.2)	0.007
No	71 (18.6)	145 (38.0)	166 (43.5)	

**Table 4 healthcare-12-02515-t004:** Differences in the mean incidence of fractures among patients according to therapy.

Type of Therapy	N	Fractures Overall		*p*-Value
		Mean	Standard Deviation	
Bisphosphonate				
Yes	47	1.0	0.6	0.617
No	384	1.0	0.6	
Denosumab				
Yes	99	0.7	0.5	0.001
No	332	1.1	0.7	
Vitamin D (cholecalciferol)				
Yes	114	1.1	0.5	0.062
No	317	1.0	0.6	
Calcium (corrected)				
Yes	148	1.1	0.5	0.072
No	283	0.9	0.6	
Vitamin D (calcitriol)				
Yes	29	1.0	0.4	0.240
No	402	1.0	0.6	
Glucocorticoids				
Yes	27	0.4	0.5	0.638
No	404	0.3	0.5	

## Data Availability

The data used and analyzed during the current study are available from the corresponding author upon reasonable request.

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
