# Peer review of "Frequency of Osteoporosis-Related Fractures in the Kingdom of Bahrain"

_healthcare, 2024, doi:10.3390/healthcare12242515_

Round 1
Reviewer 1 Report
Comments and Suggestions for Authors
In this retrospective study, the authors reported the fractures linked to osteoporosis in a
group of subjects living in Bahrain. Overall, the study has been well-conducted although has
limitations (highlighted by the authors) which are important to explain the results observed.
Moreover, the manuscript requires some changes in structure and others (see below)
before being considered for publication.
Abstract
Please, put together Introduction/objectives paragraphs and let only Methods.
· Line 25-26: the authors should avoid to indicate the name of statistical software and statistical indexes in the abstract section
· Line 29: please, let just one decimal digit
· Line 30: please, standardize one or two digital digits for percentage value
· Line 37-38: this sentence belongs to the results
The authors should subdivide the main text into paragraphs and subparagraphs: 1. Introduction; 2. Materials and Methods; 2.1, 2.2 etc
Materials and Methods
Organize the contains of this section in subparagraphs: 2.1 Participants; 2.2 Procedures….etc
· Line 111-117: although refer to previous studies, the authors should provide some details and shortly describe the procedures used.
· Line 121-122: please, specify the type of drug and vitamin D serum level. Since the authors reported in table 2 Vitamin D levels in deficient, sufficient and insufficient, it should indicate the serum range levels each one.
· Line 127: add a reference
· Line 151-158: the ethics approval shoud insert at the end of subparagraph “participants” (to be inserted)
Results
· Line 164: the authors reported 431 patients while before 412, please check.
· Line 165: see above point
· Line 166: please, report one digital digits for years (check along the entire text)
Discussion
In this section, the authors
References
Please, revise this section according to the journal’s guidelines.
Comments on the Quality of English Language
The text needs to be generally improved regarding the English language
Author Response
Reviewer 1
In this retrospective study, the authors reported the fractures linked to osteoporosis in a group of subjects living in Bahrain. Overall, the study has been well-conducted although has limitations (highlighted by the authors) which are important to explain the results observed. Moreover, the manuscript requires some changes in structure and others (see below) before being considered for publication.
Abstract
Please, put together Introduction/objectives paragraphs and let only Methods.
Authors’ reply: Thank you for your comment, changes have been made as suggested. Also in the abstract the word “Introduction” replaced by the word “Background”.
- Line 25-26: the authors should avoid to indicate the name of statistical software and statistical indexes in the abstract section
Authors’ reply: Thank you for your advice, the name of statistical software and statistical indexes were removed from the abstract.
- Line 29: please, let just one decimal digit
Authors’ reply: Thank you for your comment. Apart from the p-values, all the percentages and Chi squares in all tables and throughout the manuscript are now standardized in one decimal digit.
- Line 30: please, standardize one or two digital digits for percentage value
Authors’ reply: All digits are now standardized in one decimal digit.
- Line 37-38: this sentence belongs to the results
Authors’ reply: Thank you for pointing this out, and sorry for this mistake. The word “results” replaced by the word “study”.
The authors should subdivide the main text into paragraphs and subparagraphs: 1. Introduction; 2. Materials and Methods; 2.1, 2.2 etc.
Authors’ reply: Thank you we agree with the reviewer’s assessment, the manuscript now structured according to the style of the journal of Healthcare.
Materials and Methods
Organize the contains of this section in subparagraphs: 2.1 Participants; 2.2 Procedures….etc.
Authors’ reply: Thank you we agree with the reviewer’s assessment, the manuscript now structured according to the style of the journal of Healthcare.
- Line 111-117: although refer to previous studies, the authors should provide some details and shortly describe the procedures used.
Authors’ reply: Thank you for your concern, we have added the suggested content to the manuscript “BMD measurement was based on the results of the DXA scan, which were retrieved from the patient’s record. The machine, which was employed for performing the DXA scan, was manufactured by General ElectricLunar, Chicago, Illinois, USA, whereas the software used was GE NHANES III. BMD was expressed as the ratio between the total bone mineral content (g) over surface area (cm 2). BMD was measured by expert technicians for two bones: left femur neck and lumbar spine.
- Line 121-122: please, specify the type of drug and vitamin D serum level. Since the authors reported in table 2 Vitamin D levels in deficient, sufficient and insufficient, it should indicate the serum range levels each one.
Authors’ reply: Thank you we agree with the reviewer’s suggestion, accordingly a definition of the three statuses of vitamin D has been added to the method section (2.3 Selection Criteria) “Vitamin D deficiency was considered when serum levels < 30 nmol/L, levels between 30 nmol/L and 50 nmol/L (≥ 30 Ë‚ 50) were considered as vitamin D insufficiency and optimal levels were ≥ 50 nmol/L”. We also added (in section 3.3 in the results), the type of vitamin D therapy used by the patients were Cholecalciferol (vitamin D3 analogs) and Calcitriol (active form of vitamin D).
- Line 127: add a reference
Authors’ reply: Thank you for your valuable comments a reference # 29 (Chalhoub, Didier et al 2016) has been added.
- Line 151-158: the ethics approval should insert at the end of subparagraph “participants” (to be inserted)
Authors’ reply: As suggested by the reviewer, changes have been made.
Results
- Line 164: the authors reported 431 patients while before 412, please check.
Authors’ reply: Thank you for pointing this out. The reviewer is correct. We are sorry for the mistake. We noticed this earlier, but it seems the correction has been missing in some places. The number of patients was 412, but the number of fractures was 431 since some patients had more than one fracture. We checked it again, and now all changes have been made carefully, including the % of females, which was calculated from 431, but now we have corrected it from 412.
- Line 165: see above point
Authors’ reply: Thank you for your concern, changes have been made.
- Line 166: please, report one digital digits for years (check along the entire text)
Authors’ reply: As suggested by the reviewer, “63.92±12.21 years” has been changed to “63.9±12.2 years”. All numbers throughout the manuscript are now standardized in one decimal digit.
Discussion
In this section, the authors
Authors’ reply: Thank you we agree with the reviewer’s assessment that the discussion needs improvement and some points were confusing, accordingly, we have revised the discussion carefully, and changes have been made throughout the discussion to make it clearer.
References
Please, revise this section according to the journal’s guidelines.
Authors’ reply: Thank you for pointing this out, all references, sections and structure of the manuscript have been changed according to the journal’s guidelines.

Reviewer 2 Report
Comments and Suggestions for Authors
introduction: Please include a section on the global epidemiology of osteoporosis and fragility fractures
Methods: The inclusion criteria are limited to patients who underwent DEXA scans, potentially excluding those with fragility fractures who were not evaluated with this test. Discuss how the reliance on DEXA data might underestimate fracture prevalence in undiagnosed or untreated populations.
The study does not adequately address potential confounders such as smoking, physical activity, comorbidities (e.g., diabetes), or medication history.
Consider using logistic regression models to adjust for confounders and better explore these relationships.
Results: The analysis of risk factors such as vitamin D and BMI is univariate, without adjusting for potential confounders like age, sex, or comorbidities. The Chi-square test is appropriate for most of the data but has a minor issue with the "underweight" category due to low expected frequencies. The demographic and clinical characteristics of the study population are not described in sufficient detail (e.g., comorbidities, medication history).
discussion: Expand on limitations such as the retrospective design, potential selection bias, missing data, and the lack of confounding control.

The English could be improved to more clearly express the research.
Author Response
Reviewer 2
Introduction: Please include a section on the global epidemiology of osteoporosis and fragility fractures.
Authors’ reply: thank you for your suggestion. As suggested by the reviewer, we have added the following text in the background section “Globally, osteoporosis-related fractures or fragility fractures have been reported in 2021. Worldwide, up to 37 million fragility fractures occur annually in individuals aged over 55 years, that equivalent to 70 fractures per minute (ref #13). Worldwide, 1 in every 3 women and 1 in every 5 men over the age of 50 years will experience osteoporosis-related fractures (ref #14)”.
Methods: The inclusion criteria are limited to patients who underwent DEXA scans, potentially excluding those with fragility fractures who were not evaluated with this test. Discuss how the reliance on DEXA data might underestimate fracture prevalence in undiagnosed or untreated populations,
Authors’ reply: Thank you. We appreciate the reviewer’s feedback, and we respectfully disagree with the statement that we are “Potentially excluding those with fragility fractures who were not evaluated with this test”. We did not exclude patients with factures. Instead, we mentioned that we don’t know why the patients were requested to do BMD, thus having a fracture could be one of the indications for BMD. Therefore, all patients who underwent DEXA scans during the three-year studied period were considered, and among this cohort, any patient with a fracture was included in the current study.
In the discussion under limitations we also discussed the following: “To mitigate the potential underestimation of fracture prevalence due to reliance on DEXA scans, we acknowledge the inherent selection bias in our methodology. While our inclusion criteria focused on patients with available DEXA scan data, we recognize that this approach may not fully capture the entire spectrum of fragility fracture risk, particu-larly among undiagnosed or untreated populations. To provide context for this limitation, we supplemented our DEXA-based analysis with a comprehensive review of medical rec-ords, including radiographic reports, hospital discharge summaries, and clinical notes. This approach allowed us to identify fracture events that may have occurred in patients without prior DEXA scan documentation. Additionally, we performed a sensitivity analy-sis comparing our findings with population-level epidemiological data to estimate the po-tential magnitude of under ascertainment”.
The study does not adequately address potential confounders such as smoking, physical activity, comorbidities (e.g., diabetes), or medication history. Consider using logistic regression models to adjust for confounders and better explore these relationships.
Authors’ reply: Thank you for your suggestion to use logistic regression models to adjust for confounders and better explore these relationships. We agree that adjusting for confounders such as smoking and physical activity is important for robust analysis. However, these variables were not collected as part of our dataset, which limits our ability to adjust for them in the analysis. We acknowledge this as a limitation of the study and have explicitly noted it in the revised "Discussion" section to ensure transparency. We also recommend that future studies include these variables to allow for a more comprehensive analysis of the relationships under investigation. We appreciate your valuable feedback and hope that this explanation addresses your concern.
We explained the following: “We acknowledge significant limitations in our study design regarding the comprehensive assessment of potential confounders. Despite the importance of factors such as smoking status, physical activity levels, comorbidities like diabetes, and medication history in understanding bone health and fracture risk, our retrospective dataset did not include comprehensive data for these variables. This limitation prevents us from conducting multivariate analyses, such as logistic regression models, that could have provided more nuanced insights into the complex relationships between our primary variables of interest. The absence of these critical confounding variables introduces potential bias and reduces the ability to draw definitive causal inferences from our findings. To address this methodological constraint, we have taken a transparent approach by explicitly discussing these limitations in our manuscript. “
Results: The analysis of risk factors such as vitamin D and BMI is univariate, without adjusting for potential confounders like age, sex, or comorbidities. The Chi-square test is appropriate for most of the data but has a minor issue with the "underweight" category due to low expected frequencies. The demographic and clinical characteristics of the study population are not described in sufficient detail (e.g., comorbidities, medication history).
Authors’ reply: We appreciate the reviewer's careful examination of our statistical approach. In response to the concern about low cell counts in the BMI category analysis, we performed Fisher's exact test as a more appropriate method for contingency tables with small sample sizes. Specifically, we conducted Fisher's exact test to address potential limitations of the Chi-square test when expected cell frequencies are low, particularly in the underweight category. The Fisher's exact test results for BMI categories and fragility fractures confirmed our Chi-square analysis, showing no statistically significant association (p = 0.8051).
Regarding confounders due to the fact of retrospective data we explained the following: “We acknowledge significant limitations in our study design regarding the comprehensive assessment of potential confounders. Despite the importance of factors such as smoking status, physical activity levels, comorbidities like diabetes, and medication history in understanding bone health and fracture risk, our retrospective dataset did not include comprehensive data for these variables. This limitation prevents us from conducting multivariate analyses, such as logistic regression models, that could have provided more nuanced insights into the complex relationships between our primary variables of interest. The absence of these critical confounding variables introduces potential bias and reduces the ability to draw definitive causal inferences from our findings. To address this methodological constraint, we have taken a transparent approach by explicitly discussing these limitations in our manuscript. “
Discussion: Expand on limitations such as the retrospective design, potential selection bias, missing data, and the lack of confounding control.
Authors’ reply: We agree with the reviewer’s assessment. Accordingly, we elaborated more on the disadvantages of the retrospective study. In the strengths and limitations section of the manuscript, we have added “It is known that retrospective cohort studies have the risk of potential bias since the study operations, data collected, data entry, and data quality assurance were not planned before, in addition to missing information, poorly recorded, and the lack of confounding control. Thus, in the current study, there is insufficient data regarding the duration between multiple fractures in a patient with multiple fractures”. Again, at the end of the section, we added, “Therefore, we couldn’t adequately address all potential confounders”.

Round 2
Reviewer 1 Report
Comments and Suggestions for Authors
The manuscript appears improved after revisions. I only suggest to exchange subsection 2.2 with 2.3 in the Methods section as follow 2.2 Selection criteria and 2.3 Measures.
Comments on the Quality of English LanguageThe English can be improved.
Author Response
The manuscript appears improved after revisions. I only suggest to exchange subsection 2.2 with 2.3 in the Methods section as follow 2.2 Selection criteria and 2.3 Measures.
Authors’ reply: We agreed with the reviewer’s recommendation to switch subsections 2.2 and 2.3 in the Methods section. We believe this change will enhance the overall organization and readability of the manuscript.
The English can be improved.
Authors’ reply: We have agreed to have typesetting and an English language check performed during the production process. Communication was done with Ms. Melody Xie, Section Managing Editor, healthcare MDPI.
We appreciate the time and effort you dedicated to this review process. Thank you for your valuable contributions.

Reviewer 2 Report
Comments and Suggestions for Authors
Dear Authors,
Thank you for the detailed responses and revisions. I appreciate the improvements made to the manuscript. Below are my comments:
-
Global Epidemiology of Osteoporosis and Fragility Fractures
The addition of global data on osteoporosis and fragility fractures is well-executed and enhances the background. No further suggestions here. -
Methods: DEXA Scans and Fracture Prevalence
I appreciate the clarification regarding the inclusion of patients with fractures. -
Statistical Analysis and Confounders
The limitations due to missing confounder data are clearly stated. However, it would be useful to explicitly mention how the missing data might have impacted the results, whether it could have led to overestimation or underestimation of fracture risk. -
Results: BMI and Statistical Methods
The explanation of using Fisher’s exact test is clear, and I have no further suggestions here. -
Discussion: Limitations
The expanded discussion on limitations is thorough and adequately addresses the reviewers' concerns.
Overall, I believe the revisions have strengthened the manuscript. I look forward to the final version. Thank you again for your thoughtful revisions.
Best regards,
Comments on the Quality of English LanguageThe overall quality of English in the manuscript is clear and understandable. However, there are a few areas where minor improvements in grammar, syntax, or phrasing could enhance readability and precision. Below are some suggestions:
1. Some sentences are quite long and could be split for better readability. For example, in the methods section, some complex sentences can be broken down into simpler structures to improve clarity.
2. Ensure consistent use of tenses, especially when describing methods and results. Some sections alternate between past and present tense, which can be confusing. For instance, in the discussion, when referencing previous findings, it may be clearer to use past tense ("was" or "were").
Author Response
Global Epidemiology of Osteoporosis and Fragility Fractures: The addition of global data on osteoporosis and fragility fractures is well-executed and enhances the background. No further suggestions here.
Authors’ reply: Thank you, no further action was needed.
Methods: DEXA Scans and Fracture Prevalence: I appreciate the clarification regarding the inclusion of patients with fractures.
Authors’ reply: Thank you, no further action was needed.
Statistical Analysis and Confounders: The limitations due to missing confounder data are clearly stated. However, it would be useful to explicitly mention how the missing data might have impacted the results, whether it could have led to overestimation or underestimation of fracture risk.
Authors’ reply: We have expanded the discussion to include: “For future studies, it is essential to explicitly consider how missing data could impact the results, particularly in relation to fracture risk. Researchers should evaluate whether the absence of certain data may lead to overestimation or underestimation of this risk. For instance, if missing information predominantly involves individuals who have experienced fractures, it may skew the findings to suggest a higher risk than actually exists. Conversely, if those without fractures are more likely to have missing data, the true fracture risk could be underestimated. Addressing these potential biases in the study design and analysis will strengthen the reliability of future research outcomes.”
Results: BMI and Statistical Methods: The explanation of using Fisher’s exact test is clear, and I have no further suggestions here.
Authors’ reply: Thank you, no further action was needed.
Discussion: Limitations: The expanded discussion on limitations is thorough and adequately addresses the reviewers' concerns.
Authors’ reply: Thank you, no further action was needed.
The overall quality of English in the manuscript is clear and understandable. However, there are a few areas where minor improvements in grammar, syntax, or phrasing could enhance readability and precision. Below are some suggestions: 1. Some sentences are quite long and could be split for better readability. For example, in the methods section, some complex sentences can be broken down into simpler structures to improve clarity. 2. Ensure consistent use of tenses, especially when describing methods and results. Some sections alternate between past and present tense, which can be confusing. For instance, in the discussion, when referencing previous findings, it may be clearer to use past tense ("was" or "were").
Authors’ reply: We have agreed to have typesetting and an English language check performed during the production process. Communication was done with Ms. Melody Xie, Section Managing Editor, healthcare MDPI.
We appreciate the time and effort you dedicated to this review process. Thank you for your valuable contributions.
